# The Keratinocyte in the Picture Cutaneous Melanoma Microenvironment

**DOI:** 10.3390/cancers16050913

**Published:** 2024-02-23

**Authors:** Ramona Marrapodi, Barbara Bellei

**Affiliations:** Laboratory of Cutaneous Physiopathology and Integrated Center of Metabolomics Research, San Gallicano Dermatological Institute, IRCCS, Via Elio Chianesi 53, 00144 Rome, Italy; ramona.marrapodi@ifo.it

**Keywords:** melanoma, tumor microenvironment, skin, keratinocytes

## Abstract

**Simple Summary:**

The tumor environment is the place where the tumor resides implying that it might evolve in the role of disease progression, particularly in the phase of tumor dissemination. For cutaneous melanoma, in the early phase of disease advancement, the microenvironment is confined to the epidermal compartment since the dermo-epidermal basement membrane prevents dermal invasion. Here, poorly aggressive melanoma cells gain the aggressiveness necessary for disease progression. After dermo-epidermal membrane breakdown, the melanoma cells progressively lose keratinocyte control and engage in unusual partnerships mainly with dermal fibroblasts.

**Abstract:**

Melanoma progression is a multistep evolution from a common melanocytic nevus through a radial superficial growth phase, the invasive vertical growth phase finally leading to metastatic dissemination into distant organs. Melanoma aggressiveness largely depends on the propensity to metastasize, which means the capacity to escape from the physiological microenvironment since tissue damage due to primary melanoma lesions is generally modest. Physiologically, epidermal melanocytes are attached to the basement membrane, and their adhesion/migration is under the control of surrounding keratinocytes. Thus, the epidermal compartment represents the first microenvironment responsible for melanoma spread. This complex process involves cell–cell contact and a broad range of secreted bioactive molecules. Invasion, or at the beginning of the microinvasion, implies the breakdown of the dermo-epidermal basement membrane followed by the migration of neoplastic melanocytic cells in the superficial papillary dermis. Correspondingly, several experimental evidences documented the structural and functional rearrangement of the entire tissue surrounding neoplasm that in some way reflects the atypia of tumor cells. Lastly, the microenvironment must support the proliferation and survival of melanocytes outside the normal epidermal–melanin units. This task presumably is mostly delegated to fibroblasts and ultimately to the self-autonomous capacity of melanoma cells. This review will discuss remodeling that occurs in the epidermis during melanoma formation as well as skin changes that occur independently of melanocytic hyperproliferation having possible pro-tumoral features.

## 1. Introduction

Melanoma is a malignant tumor, caused by the transformation of melanocytes responsible for melanin production [1]. Melanoma represents only 1% of skin cancers but it accounts for most deaths due to cutaneous neoplasms [2]. Its incidence has increased in the last decade especially in the Caucasian population since it affects fair-skinned people with blond, red hair and light-colored eyes more frequently [3,4]. Furthermore, according to the Global Cancer Observatory, there were approximately 325,000 new cases of melanoma worldwide in 2020, of which about 150,000 were diagnosed in Europe [5]. The overall incidence is slightly higher in men than in women with Age Standardized Incidence Rates of 3.8/100,000 and 3.0/100,000 respectively [5]. The average age of melanoma diagnosis is 57 years. Between the ages of 25 and 40, the incidence is higher among women, but after 75 years of age, the incidence for men is 3 times higher than for women [6]. Although melanoma is considered a multifactorial disease, prolonged and unprotected natural sunlight exposure or indoor UV lamps are recognized as the main causes of disease onset [7,8]. On the Earth’s surface, UV rays are composed of UVA and UVB since UVC is absorbed by the ozone layer. UVB is the main cause of sunburn. UVB, having a shortened wavelength, directly causes DNA damage, while UVA, having a longer wavelength, acts indirectly by inducing the generation of reactive oxygen species (ROS) responsible for lipid peroxidation, protein carbonylation, and DNA lesions [9,10,11]. The consequence of intracellular signaling alteration contributes to dermis remodeling, inflammation, hyperpigmentation, and acceleration of the aging process [10]. UV-caused damage is cumulative and results in a collection of random unrepaired mutations that when involved in genes implicated in cell cycle regulation might cause hyperproliferation of skin cells. Indeed, in line with the general notion that old organisms are more sensitive to stress, the capacity to repair DNA defects is subject to age-dependent decline [12]. Accordingly, the number of somatic mutations found in normal human tissue as well as in tumors increases with age [13]. On the other hand, the accumulation of mutations may contribute to aging and apoptosis perturbing the physiological balance existing between cell death and cell renewal and exacerbating tissue functional decline [14]. Several studies evidenced that a family history of skin cancers is strongly associated with an increased risk of melanoma in both sexes [15]. A history of non-melanoma skin cancer increases the risk of melanoma and also in the skin of color subjects having a generally lower incidence of skin cancer [16]. A number of phenotypic traits connected to pigmentation portend increased risk for melanoma. People with fairer skin phenotypes (pale white skin, light-colored eyes, low tanning ability, burning propensity, high freckling tendency) are more likely to develop cutaneous melanoma [9,17] as well as people habitually exposed to the sun for prolonged periods [18]. Family history is one of the most significant risk factors for melanoma [9]. Approximately 8–12% of melanoma cases occur in multiple-case families and can be inherited in an autosomal dominant fashion [19]. Interestingly, risk based on family history depends not only on the number of individuals in the family who experience melanoma but also on the number of melanomas in each family member [20]. Germline mutations of *CDKN2A*, *CDK4*, *TERT*, and *POT1* genes have been identified as high-risk factors for melanoma susceptibility [21,22,23], whereas *MC1R* (melanocortin 1 receptor) polymorphic variants are reported in 60.5–82.1% of melanoma [24,25]. Currently, technological advances have allowed the identification of additional genes involved in melanoma susceptibility: *BRCA2*, *BAP1*, and *MITF* [26]. Also, pathogenic variants in albinism genes may account for a minor part of familiar melanoma [27]. More recently, *CDH23*, *ARHGEF40*, *BRD9* [28], and *CASP8* [29] have been proposed as candidate susceptibility genes in hereditary melanoma [30,31]. However, in general, it is important to take into consideration that inhered cancer-associated gene variants might influence the biology of all cell types including those of the tumor microenvironment augmenting or reducing the personal oncologic risk. Common variants of the *MC1R* gene play a relevant role also in sporadic melanoma onset [32,33]. Engagement of the MC1R expressed on the cell membrane by agonists activates the cAMP pathway leading to pro-melanogenic gene transcription and enhancement of DNA repair and antioxidant capacity [34]. This is explained by the evidence that several *MC1R* polymorphisms are associated with light-skin phenotype and attenuated hormonal stimulation of the melanogenic biosynthetic pathway [35]. Consistently, evolutionary constraints have determined the geographical distribution of undermelanized variants. MC1R nonsynonymous variants are largely present in Asian and European populations but not in African ones [36,37]. However, increased melanoma risk for individuals carrying polymorphic forms of *MC1R* is not restricted to cAMP signaling and resultant weakened pigment-dependent function (prevalent synthesis of reddish pheomelanin having photosensitizer properties and insufficient UV-ray absorption), but is additionally attributed to reduced protection against oxidative damage, attenuated MAPKs (mitogen-activated protein kinases), AKT (also known as protein kinase B, PKB) and PI3K (phosphoinositide 3-kinase) signaling activation [38,39,40,41,42], immune responses, glucose metabolism, and energy homeostasis modulation [43]. The increased risk associated with *MC1R* variants spans all histological subtypes and it occurs on both chronically and intermittently sun-exposed skin [44]. *MC1R* genetic variants are determinant also in the acquisition of skin aging-related features [45,46] including specific dermal variation during photoaging [47,48]. Since properties referred to functional aging of the skin have been associated with the potential pro-tumorigenic phenotype of fibroblast [49,50], impaired functionality of the (α-melanocyte stimulating hormone) α-MSH/MC1R axis might contribute to creating a favorable milieu for hyperproliferative diseases in photo-exposed bodies’ area. Notably, in the skin, the features of senescent fibroblast largely overlap with those of fibroblast associated with neoplastic cells [51].

MC1R signaling favors tolerance to environmental stressors by increasing the capacity to repair DNA damage for genomic maintenance and cell survival [40,52,53]. Consequently, a higher somatic mutation burden has been demonstrated in melanoma from subjects with *MC1R* variants compared to individuals with the wild-type sequence [54]. Accordingly, independent of skin type and hair color, some genetic variants of *MC1R* confer susceptibility also to non-melanoma (basal cell carcinoma, BCC, and squamous cell carcinoma, SCC) skin cancer [32,55]. MC1R activation lessens T-cell infiltration in the tumor microenvironment and causes a melanoma-specific mechanism of checkpoint blockade therapy resistance and immune evasion [56]. Moreover, MC1R variants correlate with poor response to BRAF inhibitors [57]. As part of the “qualitative” consideration concerning non-synonymous genomic variants, very recently, high MC1R levels of expression in primary and metastatic melanoma have been associated with poor prognosis [58]. Inverting the viewpoint, paradoxically a possible explanation for this observation might reside in the overexpression of DNA repair and anti-apoptosis pathways that support the persistence of highly injured cells [59].

Concerning the identification of melanoma risk factors, awareness is also addressed to melanocytic nevi that represent a benign lesion resulting from an abnormal proliferation of melanocytes. Since nearly 25% of melanomas appear in pre-existing nevi [60,61], an elevated number of nevi and the presence of atypical nevi constitute an important risk factor for the onset of melanoma [62]. Melanocytic nevi are precancerous lesions resulting from a focal and limited proliferation of melanocytes driven by the mutational activation of an oncogene such as BRAF [63]. After forming a nevus, nevomelanocytes undergo an enduring exit from the cell cycle that prevents them from progressing to melanoma. Melanoma can also originate from lentigo maligna, a slowly growing lesion in situ, which generally arises in the face of elderly people with photodamaged skin [62]. It has been estimated that 2–5% of lentigo maligna progress to invasive melanoma [64] and that lentigo maligna cover about 10% of all melanomas [65]. Desmoplastic melanoma partially overlaps with lentigo maligna due to the common link to sun-damaged skin but presents a distinct genetic profile [66].

In the context of cancer, the tissue surrounding pathologic cells acquired more and more interest.

During its evolution, melanoma progressively shapes host tissue into a favorable milieu promoting cancer dissemination, resistance to therapy, and immune evasion [37,67,68]. However, some inherited or acquired characteristics including chronic inflammation, fibrosis, and immunosuppression might predispose to hyperproliferative diseases [69,70]. Current studies have highlighted the role of aging in shaping the skin into a pro-tumorigenic condition [51,71,72,73]. Correspondingly, the incidence of tumors positively correlates with age [74] and progeroid syndrome, a heritable autosomal recessive human disorder characterized by the premature onset of numerous age-related diseases, predisposing to cancers [75]. Properly, skin cancers are the most common malignancies affecting older adults [76]. While in normal aging, the acquisition of cell-senescent phenotype and functional decline over time acts synchronously in various tissues, premature senescence may be limited to one or a few organs such as the skin since it is continuously exposed to external aggression. Extrinsic aging is superimposed on intrinsic aging resulting in a more intense rate of senescent cell accumulation than in normal aging. In neoplasms, crosstalk between melanoma and dermal fibroblasts also referred to as cancer-associated fibroblasts (CAFs) or more precisely melanoma-associated fibroblasts (MAFs) plays a relevant role in tumor growth, invasive abilities, and resistance to therapy [37,77,78]. Other cutaneous cells are involved in melanocyte transformation into melanoma and disease progression. Among these, keratinocytes forming the multilayer outer skin compartment, being in direct contact with neoplastic cells in the melanoma niche are deeply implicated in melanomagenesis, especially during early phases. However, compared to other components of the melanoma stroma, including mesenchymal and immune cells (natural killer cells, T lymphocytes, B-lymphocytes, dendritic cells, macrophages, and adipocytes), keratinocytes have been less investigated yet. This review will present the current knowledge concerning the direct and paracrine communication between keratinocytes and melanocytes in melanomagenesis.

## 2. Material and Methods

This manuscript considered data from reviews and experimental in vitro as well as in vivo studies with the thematic of melanoma and keratinocyte (epidermis) with a specific emphasis on articles comprising both topics. This analysis covers a multitude of recent and current peer-reviewed manuscripts without any restriction related to the publication period. Since this review has been designed as a narrative review, it synthesizes without systematic methods.

## 3. Skin Physiology

### 3.1. Skin Structure

Human skin covering the entire body acts as a barrier preventing the loss of water and protecting it from physical, chemical, and microbial damage. It is structured in three layers: epidermis, dermis, and hypodermis. The epidermis is mainly composed of keratinocytes, sparse immune cells, and melanocytes allocated at the base of the epidermis (Figure 1A).

The dermis is the middle layer of the skin. It confers flexibility and strength to the cutis due to the abundance of well-organized extracellular matrix proteins (collagen and elastin, glycoproteins, and glycosaminoglycans) [79,80]. Even if cellularity is modest in the dermal compartment, it hosts the highest functional and morphological cell diversity [81]. In addition to sparse fibroblasts, the most abundant cell type, the dermis, contains mesenchymal stem cells, lymphocytes, natural killer cells, Langerhans cells, Merkel cells, macrophages, mast cells, and dendritic cells [82]. The dermis also encloses blood and lymph vessels, sweat and sebaceous glands, hair follicles, nerves, and a variety of sensory nerve receptors. The subcutaneous fat layer (or hypodermis) is predominantly composed of adipocytes, pre-adipocytes, and mesenchymal stem cells but also of fibroblasts, pericytes, macrophages, T cells, and erythrocytes in a stromal network of collagen, vascular elements, nerve endings, and muscles [83]. The primary function of the hypodermis is to regulate the body’s temperature, and storage energy, and protect from injury underlying structures such as muscle and bones. Recent work has demonstrated a remarkable plasticity of adipocyte-lineage cells (stem cells and mature cells) during tissue repair indicating a central role of these cells in normal and pathological tissue remodeling. Further, due to the dynamic nature of adipocytes as they deliver bioactive products, lipids, and adipokines, hypodermic fat is emerging as an essential regulator of several physiological skin processes such as wound healing, pigmentation, hair growth, and innate immune response [84,85]. More, some evidence argues for the role of subcutaneous adipocytes in skin tumorigenesis [86,87]. Adipocyte-derived lipids represent a potent energy source capable of increasing melanoma proliferation and invasion in vivo and in vitro models [86,88]. Transferring high amounts of fatty acids, adipocytes alter the mitochondria metabolism of adjacent cells rendering melanoma cells less dependent on de novo intrinsic lipogenesis and accelerating melanoma development in the zebrafish model [88]. Since to be in direct contact with adipocytes, melanoma cells require a breakdown of the dermal-epidermal junction of the skin and the gain of the vertical growth phase, adipocytes are likely important for the metastatic process in melanoma. Lastly, adipocytes are a local source of CAF [89] since this type of cell might phenotypically change to generate fibroblast-like cells, termed adipose-derived fibroblasts, expressing some specific CAF markers and a marked pro-inflammatory phenotype [90,91].

### 3.2. Epidermal Organization

Epidermal cells are organized in a multilayered stratified structure prevalently occupied by keratinocytes at different levels of differentiation [92]. Epidermal thickness varies depending on the anatomical location and the presence of the appendix. The basal membrane, also known as basal laminae, provides the primary function of anchoring down the basal keratinocytes of the epidermis to the connective tissue of the papillary dermis. Keratinocytes are anchored to the basal membrane through hemidesmosomes and to adjacent cells with desmosomes (specialized adhesion structures) [92,93]. In healthy skin, melanocytes are confined in the basal layer. As discussed below, passing the basal membrane and invading the dermis is a crucial step for the acquisition of metastatic competence of melanoma cells [37].

Keratinocytes modify their appearance from one layer to the next with the function of completing a convoluted differentiation process. Starting from the bottom of the epidermis upwards, we find the stratum basale, stratum spinosum, stratum granulosum, and stratum corneum. Stratum basale is a germinative layer and it is the deepest layer of the epidermis. It comprises a monolayer of keratinocytes with high mitotic activity, responsible for the intrinsic continuous renewal of the skin. Here, the self-renewing cell population (stem cells) ensures continuous epidermal renewal in the course of the human lifetime. During the migration, cells gradually fill the cytoplasm with intermediate filament proteins (keratins 1 and 10 and filaggrin) and lipids which conifer toughness to the epidermis [92]. At the same time, keratinocytes change architecture from cuboidal to polygonal shape that progresses through largely flattered morphology culminating in the disk-like shape of corneocytes (also called squamous cells). As part of the differentiation program, keratinocytes lose their nucleus and the major part of cytoplasmic organelles through a degenerative evolution partially overlapping those of the apoptotic process [94,95]. Finally, to complete their differentiation (cornification), a cornified envelope replaces the plasma membrane and keratinocytes undergo a specific pH- and calcium-dependent cell death (corneoptosis) [96]. Indeed, the stratum corneum is composed of 20–30 layers of corneocytes filled with large granules of keratohyalin of which the major constituent is pro-filaggrin [97]. Within the cells, there are also lamellar bodies, consisting of lipids (mainly glycolipids, cholesterol, and ceramides); the lipids released into the extracellular space through exocytosis form a hydrophilic barrier on the corneocytes (corneocyte lipid envelope) and confer waterproofness to the cutis [98]. The protein fraction of the corneocyte envelope includes proteins like involucrin, loricrin, envoplakin, and periplakin cross-linked to lipids [96]. Stratum corneum represents the first line of defense against several insults, protecting the skin from UV-irradiation and chemical injury, preventing toxins and microorganisms from entering the body, and playing a crucial role in trans-epidermal water loss [99]. Corneocytes are held together through corner-desmosomes until cells leave the stratum corneum by desquamation, the process of cell shedding from the skin’s surface [92].

The epidermis hosts a relevant number of resident immune cells and derived molecular mediators. Leucocytes, dendritic T cells, macrophages, mast cells, natural killer, and Langerhans cells are strategically disseminated along all epidermal layers [100,101]. Stratum corneum includes several microbial sensors such as Toll-like receptors, antimicrobial peptides, and cytokines produced physiologically and in response to microbial invasion [102].

### 3.3. Epidermal–Melanin Units

In normal epidermis, the structural and functional architecture enclosing keratinocytes and melanocytes in an intimate relationship with each other is referred to as an “epidermal–melanin unit” (Figure 1B). The epidermal–melanin unit denotes the symbiotic relationship in which one melanocyte has the task of carrying melanosomes to neighboring keratinocytes. The transfer of melanin protects keratinocytes from UV-induced nuclear damage, which in turn controls the proliferation and differentiation of melanocytes in the basal layer of the epidermis [103]. The number of melanocytes, per unit area of the epidermis is the same in all the human skin phenotypes, hence the skin color is a function of the quantity and the type of melanin produced, the number, and size of melanosomes transferred into the surrounding keratinocytes [104,105]. Caucasians have small melanosomes, lightly pigmented usually absent in the most superficial layers of the epidermis, while in black subjects melanosomes are larger, darker, and almost reach the stratum corneum [106,107]. According to Iozumi K. et al. (1993), differences between white-skin melanocytes and dark-skin melanocytes are related to the intensity of tyrosinase activity [108]. Furthermore, the quantity and quality of melanin are influenced by a variety of genes including *MC1R*, *OCA2*, *DRD2*, *EGFR*, *DCT*, *KITLG*, *SLC24A5*, *SLC24A5*, *TYRP1*, *DTNBP1*, *MYO5A*, *MFSD1*, *DDB1*, and *HERC2*, having different impacts in the function of the ethnicity [36,105].

Hyperpigmentation can occur pathologically, as in cutaneous melasma due to an increase in melanin production and transfer with a change in the density of melanocytes resulting in dark patches on the skin, in nevi and melanoma due to benign and malignant hyperproliferation of melanin-producing cells respectively. On the other hand, the whole absence of pigmentation results in albinism, a genetic condition, and the partial absence in skin patch results in vitiligo (acquired condition) and piebaldism (genetic disorder). Post-inflammatory dermal and epidermal hyperpigmentation as well as permanent hypopigmentation are also frequently observed in human skin [109]. From the functional point of view, keratinocytes of the epidermal–melanin unit elaborate several intrinsic and extrinsic stimuli and converge them in a mechanism appropriated to support melanocyte homeostasis and melanogenesis. The transportation of melanosomes, specialized lysosome-derived organelles, where melanin synthesis and packaging occur, is allowed by dendritic protrusions of epidermal melanocytes, which can reach the upper layers of the epidermis. The mechanism of the transfer process of melanosomes from melanocytes to keratinocytes has not been completely defined. Four mechanisms of melanosome transfer have been suggested: (i) direct inoculation of melanosomes into the cytoplasm of keratinocytes via keratinocyte–melanocyte membrane fusion; (ii) release of single melanosomes from melanocytes and their subsequent endocytosis by keratinocytes; (iii) the shedding of melanosome-laden globules by melanocytes, and (iv) direct cytophagocytosis of melanocyte dendrite tips containing melanosomes by adjacent keratinocytes [110,111]. Once incorporated into recipient keratinocytes, melanosomes are predominantly arranged in a supranuclear cap and degraded as the keratinocytes undergo terminal differentiation and desquamation. In dark-skinned individuals, large melanosomes are maintained as individual organelles throughout the cytosol of the keratinocyte. In contrast, in light-skinned individuals, melanosomes are smaller and are aggregated in clusters of 4–8 units [112].

Skin melanocytes are considered intermittent mitotic cells since this type of cell proliferates on demand in case of UV exposure or regenerative processes [113]. Similar to most of the human cell types, an inverse relationship between age and the proliferative activity of melanocytes has been observed [114,115]. Melanosome transfer mainly involves the epidermal basal layer to ensure effective photoprotection for progenitor keratinocytes and stem cells that reside in the microenvironment of the basal epidermis. In the skin, the major types of melanin are brown-black eumelanin and yellow-red pheomelanin. Eumelanin is characterized by high protective properties against DNA damage induced by UV radiation, whereas pheomelanin pigment exhibits a phototoxic pro-oxidant behavior [116]. Tyrosine, a nonessential amino acid, serves as the precursor molecule for the production of all types of melanins. The key enzymes involved in this process include tyrosinase, which catalyzes the initial step of converting tyrosine into dopaquinone, tyrosinase-related protein 1 (TRP1), and dopachrome tautomerase (DCT), which further serves for eumelanin biosynthesis. Independently of the presence of TRP1 and DCT, in the presence of cysteine, dopaquinone is converted into cysteinyldopa and then into pheomelanin by oxidation. Thus, the function of the relative expression of these three enzymes depends on the ratio of eumelanin and pheomelanin and ultimately the skin color. Independently of the presence of TRP1 and DCT after conjugation with cysteine, dopaquinone is converted into pheomelanin [117]. The gene expression of Tyrosinase, TRP1, and DCT, is under the control of MITF (microphthalmia-associated transcription factor), the key transcription factor in melanocyte differentiation. MITF also enhances melanogenesis, activating the transcription of *PMEL* (a melanosome protein also reported as PMEL-17, gp100, ME20, HMB-45, or silver protein), *GPR143* (G protein-coupled receptor 143), and *MLANA* (Melan-a), coding for structural melanosome-associated proteins [118,119]. MITF is mainly regulated by cyclic adenosine monophosphate (cAMP)/protein kinase A (PKA), cAMP-response element binding protein (CREB) transcription factor, and Wnt/β-catenin signal pathways may function as an oncogene, reinforcing the existing link between melanogenesis and melanomagenesis [120,121]. MITF function is additionally controlled at post-transcriptional and post-translational levels and the availability of transcriptional partners [122]. Excluding mutations and other genetic alterations occurring in melanoma cells, the elevated number of intrinsic and microenvironmental factors contributing to MITF function modulation explains the intra- and inter-tumoral remarkable heterogeneity reported [123,124].

### 3.4. Physiological Keratinocyte-Melanocyte Communication

The interaction between keratinocytes and melanocytes is relevant to provide the homeostasis of the epidermis. First, to ensure functional optimization of the epidermal–melanin unit, keratinocytes and melanocytes are precisely organized in the space; one post-mitotic melanocyte is in contact with their dendrites with about 36–40 keratinocytes [125]. Keratinocyte–melanocyte interaction occurs through direct cell–cell physical contact as well as through intense bidirectional paracrine communication.

#### 3.4.1. Cell–Cell Contact in Melanocyte-Keratinocyte Interaction

Cell adhesion molecules such as E (epithelial)-cadherin, P (placental)-cadherin, and Desmocollin-1 are important regulators for keratinocyte differentiation by modulating homotypic and heterotypic cell–cell interactions [126,127]. Furthermore, E-cadherin is not only critical to balance the differentiation/proliferation states of keratinocytes and melanocytes but also to ensure epidermal cell survival [128]. Cadherins are transmembrane proteins included in the adherens junction with multiple Ca^2+^-dependent extracellular domains and a relatively short cytoplasmic domain that connects them to the cytoskeleton [128]. Keratinocytes and melanocytes express on the surface both E-cadherin and P-cadherin but E-cadherin is primarily responsible for the adhesion of human melanocytes to keratinocytes [129]. The membrane expression of E-cadherin and other filopodia-associated proteins is important for the transfer of melanosomes from producing cells to keratinocytes following UV exposure [130,131]. In partial contradiction to the pro-melanogenic role of cadherin-mediated adhesion, UV irradiation enhancing ET-1 (endothelin-1) release by keratinocytes downregulates E-cadherin in melanocytes and melanoma cells [132]. Recently, Pmel17 delivery in the extracellular environment by melanocytes has been described as an additional UV-induced mechanism for temporally weak melanocyte–keratinocyte contact [133]. In this study, Hu and co-authors demonstrated that a soluble form of Pmel17 when internalized in the keratinocyte interacts with the scaffold protein FHL2 (four-and-a-half LIM-domain protein 2) triggering actin cytoskeleton remodeling and E-cadherin reduction, that compressively favor melanocyte migration. However, normally this mechanism is part of the formation of a new epidermal–melanin unit that imposes the rapid recoupling with keratinocytes [134]. Dendrites retraction and melanocyte decoupling from neighboring keratinocytes is a prerequisite for melanin-producing cell division. Impairment of this process might facilitate melanoma development. The intracellular domain of E-cadherin binds β-catenin limiting the nuclear localization and function of this co-transcription factor. On the other side, nuclear β-catenin when complexed with LEF-1 represses E-cadherin transcription reinforcing the loss of adhesion [135]. GPNMB (Glycoprotein nonmetastatic melanoma protein B), another MITF-regulated melanosome-associated glycoprotein, also exists in a soluble form [136]. GPNMB also functions as an adhesion protein between basal keratinocytes and melanocytes since its extracellular domain binds to integrin in the process of maintaining cell–cell adhesion in a calcium-dependent way [137]. Both keratinocytes and melanocytes express and secrete GPNMB and the amount of this protein is augmented after UVA irradiation, by α-MSH (α-melanocyte stimulating hormone) and pro-inflammatory cytokines like IFN-γ and TNF-α. Interestingly, lack of GPNMB expression is a specific characteristic of depigmented vitiligo skin [138]. Evidence that GPNMB protects melanocytes from cytotoxicity induced by oxidative stress further argues for a role in the vitiligo pathogenic mechanism [139]. Recently, a pro-metastatic function has been attributed to the melanoma-derived soluble form of GPNMB due to its capacity to exclude T-lymphocytes from the pre-metastatic niches [136].

Desmosomes are formed by glycoproteins such as desmogleins (or desmosomal cadherins) and desmocollins. Desmoglein 1 (Dsg1), in addition to its adhesive function, can modulate keratinocytes–melanocytes paracrine communication. In particular, it has been demonstrated that melanocytes within a Dsg1-deficient 3D human skin exhibited increased pigmentation, increased production of paracrine factors such as cytokines/chemokines, and altered dendritic protrusions. Therefore, this scenario highlights that Dsg1 deficiency may contribute to pagetoid behavior, such as occurring in early melanoma development [140]. Different types of cell–cell adhesion molecules are connexin, integral membrane proteins clustered in the gap junctions, and occludins and claudins included in the tight junctions. Tight junctions allow passage of ions and small molecules between adjacent cells. Gap junctions are channels in which secondary messengers and metabolic products are easily exchanged between coupled cells [141]. Alteration of gap junction communication mostly reflects an atypical cadherin expression profile [142]. It has been demonstrated that blocking gap junctions in a co-culture system of human keratinocytes and melanocytes reduces the expression of key regulatory genes of melanogenesis such as tyrosinase and MITF in a dose-dependent manner [143]. UVB and UVA transiently alter the integrity of epidermal gap junctions and cancer cells frequently present downregulated levels of gap junctions suggesting a role of these complexes in the regulation of UV-induced inflammation and skin carcinogenesis [131,144].

In normal epidermis, integrins are important for the connection of basal cells to extracellular matrix components such as collagen, laminin, and fibronectin and for any type of tissue remodeling including normal migration of cells and wound healing. Alpha2/beta 2 integrins are implicated in resident lymphocyte recruitment due to the binding to ligands of the Ig immunoglobulin family, such as ICAM-1, ICAM-2, and ICAM-3 [145].

#### 3.4.2. Cross-Talk in the Epidermal–Melanin Unit

The most evident exchange between cells in the epidermis is the melanin-containing melanosome transfer that aims to protect keratinocytes from the deleterious effect of UV irradiation [146]. Keratinocytes, on the other hand, control the proliferation, adhesion, migration, and differentiation of melanocytes (including the melanin biosynthetic pathway) through the secretion of growth factors and cytokines in a paracrine manner [147,148]. Several molecules are identified as involved in melanocyte growth, such as basic fibroblast growth factor (bFGF), stem cell factor (SCF), endothelin 1 (ET1), ET3, and hepatocyte growth factor (HGF) [149,150]. Among these mitogenic factors, ETs, SCF, and HGF are mainly provided by keratinocytes. In physiological conditions, the production of POMC, the α-MSH precursor, bFGF, and ET-1 is mostly due to p53-dependent transcriptional activation in response to UV [40,151]. These mitogens display a synergistic effect on melanocytes, activating the MAP Kinase pathway, especially the kinase ERK2 [152]. P53 plays a central role in hyperpigmentation [153]. At variance, mutations in the p53 gene are rare in primary melanoma [154]. In contrast, repeated daily exposure to UVB induces a p53-dependent melanocyte senescence [155]. Keratinocytes release a significant amount of bFGF after UV irradiation increasing expression of focal adhesion kinase p125^FAK^ on melanocytes and facilitating their migration [156]. Transwell-based assay of normal human epidermal melanocytes demonstrated that PI3K/Akt-Rac1-FAK-JNK-ERK signaling pathways lead to cytoskeleton reorganization contributing to melanocyte migration after bFGF stimulation [157]. Similarly, HGF-binding transmembrane tyrosine kinase receptor-c-Met stimulates NF-kB and MAPK increasing the proliferation and motility of melanocytes [158]. Upon ligand binding, c-Met activates PI3K MAP Kinases through the Gab1/Grb2-SOS Ras pathway [159,160]. After UVB irradiation, epidermal cells strongly induce HGF expression. Further, keratinocytes produce a large amount of IL-1α that stimulates HGF production by dermal fibroblasts. At the same time, UVA directly upregulates HGF mRNA and protein [161].

SCF/KIT pathway is important for normal melanocytic function; experiments carried out with adult human skin xenografts injected with recombinant human SCF or a KIT-inhibitory antibody demonstrated that SCF/KIT pathway impairment elicited a melanocyte loss and a reduced expression of melanocytic differentiation antigens TRP-1 and pmel17. Hence, this investigation suggests that the SCF/KIT pathway plays a critical role in the control of human melanocyte homeostasis [162]. Keratinocytes-derived SCF has pleiotropic effects. It can modulate integrin expression at the protein level through SCF facilitating melanocyte attachment and migration on ECM ligands, a process needed during embryogenesis and the repigmentation process [163].

In the presence of IL-1α and after UVB exposure, keratinocytes produce a high amount of ET-1 triggering a stimulatory effect on DNA synthesis and melanization in cultured human melanocytes [164]. The signaling mechanism involved is represented by the crosstalk between PKC, cyclin AMP system, and MAP kinase [165]. A recent study demonstrated that after UV irradiation, ET-1 protects human melanocytes, inhibiting apoptosis and reducing DNA photoproducts in a dose-dependent manner. ET-1 enhances the phosphorylation of JNK and p38 that target the transcription factor ATF-2. ET-1 synergistically with α-MSH regulates DNA repair to in response to UV. After binding to the receptor, ET-1 causes phosphorylation of repair-DNA sensors ATM/ATR, therefore enhancing the level of DNA damage recognition proteins like XPC, VCP, XPA, and γH2AX [52].

Cell migration is essential in inflammatory reactions, remodeling, and healing. Keratinocytes express several MMPs (matrix metalloproteinases) that can cleave components of the extracellular matrix, preparing an optimal context for cell mobilization. However, normally the expression of MMPs and ADAMs (a disintegrin and metalloproteinase) are very low and their function is regulated by proteolytic activation to ensure fast remodeling of the dermis, as it became necessary. Several MMPs are UV-responsive genes. On keratinocytes, UVB activates the transcription of a cluster of MMP genes indicating that coordinated regulation of these proteins is important for damage response [166]. Specifically, MMP9 and MMP2 secreted in response to UV or proinflammatory cytokines such as IFN-γ, TNF-α, IL-1β, and IL-6 are involved in inflammatory reactions, remodeling, and healing [167,168]. In this process, keratinocytes have a direct as well as an indirect role since UVB-irradiated keratinocytes releasing IL-1 activate the expression of MMP1 in dermal fibroblasts [169]. On the other hand, MMP1 in addition to collagen damage exacerbates the skin inflammatory response [170]. Dysregulation of MMPs affects skin homeostasis by premature aging and facilitating disease development. ADAM12 is upregulated in the non-healing edge of chronic skin ulcers [171], whereas ADAM10, ADAM12, and ADAM17 [172], MMP2, MMP9, MM12, and MMP7 [173,174] are highly expressed in psoriasis, a common hyperproliferative and inflammatory skin condition. MMP9 produced by keratinocytes in response to IFN-γ has been also implicated in melanocyte detachment from the basal epidermal layer in vitiligo skin [175]. Notably, since the ability of melanocyte migration during re-pigmentation also largely depends on MMP9 activity [176], impaired MMP activity has been proposed to explain the reduced migration of melanocyte precursor from the outer root sheath in vitiligo skin [177].

## 4. Remodeling of Epidermal Microenvironment during Melanoma Onset and Progression

To replace old cells, and repair or expand the skin, melanocytes need to retract their dendrites, separate from the basement membrane and neighboring keratinocytes, proliferate, and migrate along the basement membrane before they reach a new epidermal–melanin unit. Since these processes are largely regulated locally by surrounding cells, it is reasonable that melanocytic hyperproliferations are due to the mistaken escape from keratinocyte control. Accordingly, in vitro, keratinocytes control proliferation to maintain the original keratinocyte/melanocyte ratio in a cell–cell direct contact-dependent manner [178]. Mechanistically, escape from keratinocyte control might be due to loss of contact with keratinocytes, modification of keratinocytes’ secretory behavior, or alternatively, the acquisition of unresponsiveness to inhibitory growth factors by melanocytes. During the initial radial growth phase, melanoma cells are still confined in the epidermal compartment. Thus, it is conceivable that poorly aggressive melanoma acquires aggressiveness being still surrounded by normal keratinocytes (Figure 2). This process is likely dependent on the modification of cell adhesion properties and the paracrine activity of keratinocytes.

### 4.1. Alteration of Melanocyte–Keratinocyte Physical Interaction during Melanoma Inception and Progression

Melanocytes and corresponding early hyperproliferative melanocytic cells are fully surrounded by keratinocyte layers that physiologically protect long-living pigment-producing cells. Thus, the simple reduction of these activities by stressed or damaged keratinocytes might represent in theory a pro-melanogenic condition. Sun exposure is the major environmental risk for epidermal homeostasis. Frequent excessive exposure to UV light is associated with alterations in the composition of the basement membrane and dermal extracellular matrix that might facilitate disease occurrence. Chronic UV irradiation decreases the expression of epidermal type VII collagen that serves for dermal–epidermal junction integrity and fibrillins in the upper dermis [179,180]. Moreover, the expression of β1 integrins that serve for the connection of basal keratinocytes to the basement membrane is severely compromised in sun-damaged skin [181]. At the same time, higher expression of involucrin in the stratum corneum and lower β1 integrins in the basal epidermis argue for impaired self-renewal and adhesive properties of UV-damaged keratinocytes [181]. In chronically sun-exposed skin, qualitative and quantitative modifications of extracellular matrix (ECM) proteins cause loss of tensile strength, increase fragility, and lessen reparative ability [73,182]. Even so, the tissue surrounding benign nevi and melanomas displays greater stiffness than normal skin evidencing that mechanical properties of the matrix have a role in melanoma initiation and control progression of the invasive process in advanced melanoma [183,184]. UV-irradiated skin produces also several MMPs, which degrade dermal collagen and elastic fibers causing an overall tissue remodeling that is part of the photoaging process [185]. Photoaging is characterized by poorly organized ECM, wrinkling, irregular pigmentation, telangiectasia, thinned or fragile skin, and dermal and epidermal atrophy. Epidermis-releasing MMPs display an accelerated melanocytic cell invasion of the dermis due to basal membrane destruction [186]. Instead, in the context of melanoma, excessive production of MMP1, MMP2, MMP9, and MMP19 has been frequently associated with invasion behavior of tumor cells [187,188]. Overall, modification of the balance MMPs/TIMPs (tissue inhibitors of MMPs) is critical for melanoma invasion [189]. MMP9 overexpression is frequently associated with aberrant MAPK pathway activation, a very frequent event in melanomagenesis [190]. Other mechanisms for MMP9 induction are the dysregulation of TGFβ [191], mutation in the *MMP9* gene [190], and epigenetic events [192]. TGFβ elevation in the tumor microenvironment is due to autocrine production by tumor cells as well as to the release by keratinocytes [193] and fibroblasts [194] subsequent to UV radiation. Furthermore, the expression of several components of ADAM family proteins is remarkably increased in melanoma cells, with significant differences between metastases and primary melanocytic lesions [195]. ECM degradation favors melanoma spreading due to the activation of proangiogenic factors in the surrounding tissue. Overexpression of ADAMs and MMPs involves melanoma peritumoral stromal fibroblasts [196]. A possible similar activation of keratinocytes surrounding melanoma has been suggested by in vitro evidence that medium conditioned by melanoma cells stimulates the production of MMP3, MMP9, and MMP14 by Hacat cells while lowering TIMP expression [197].

Cell adhesion molecules including cadherin, integrin, immunoglobulin, and selectin have been related to the melanoma metastatic process [128]. E-cadherin is the most critical molecule for keratinocyte–melanocyte interaction because melanocytes not expressing E-cadherin lose most of the physiological regulation by keratinocytes [198].

From the early stages of melanoma, loss of E-cadherin expression and acquisition of N (neuronal)-cadherin expression allows the melanoma cells to preferentially bind to fibroblasts that abundantly express this cadherin, and thus promote invasion into the dermis [198]. E-cadherin/N-cadherin shift represents the hallmark of epithelial-to-mesenchymal transition (Figure 3).

Distraction of E-cadherin from the membrane surface has also been implicated in the regulation of Wnt/β-catenin since its intracellular domain binds β-catenin, counteracting the nuclear localization of this key co-transcription factor for melanocyte biology. Β-catenin membrane-to-nuclear translocation and gain of nuclear localization enable the binding to members of lymphoid enhancer-binding factor (LEF)/T-cell specific factor (TCF) family and some other co-regulators to promote the transcription of ubiquitous genes (*Jun*, *c-Myc* and *CyclinD-1*) [199] and melanocyte-lineage restricted genes including *MITF* [200], *DCT*, and *Brn-2* [201,202,203]. Some recent data suggested that stromal cells might cause atypical melanocyte detachment and exclusion from the normal epidermal–melanin unit. Using a trans-well co-culture system to keep in touch with melanoma cell lines with poor aggressive behavior and healthy epidermal keratinocytes, Untiveros and collaborators demonstrated that tumor cells progressively assume a mesenchymal-like phenotype resulting in a decrease in E-cadherin expression and activating stabilization of β-catenin [204]. Similar results were recorded with dermal fibroblasts with the difference that keratinocytes additionally activated ERK signaling [204]. Discrepancies emerged comparing in vivo and in vitro studies regarding the effect of Wnt/β-catenin signaling on proliferation [205,206], invasion [202,207], and migration [208] of melanoma cells. Further, in vivo, observations demonstrated that β-catenin nuclear localization correlates with improved patient survival [208,209]. Following the loss of E-cadherin, the dropping of Dgs1 expression in keratinocytes discontinues control of melanocytes contributing to melanomagenesis [140]. Melanocytes in Dsg1-silenced skin equivalents delocalize suprabasally, contributing to the pagetoid behavior occurring in early melanoma development. In a co-culture model, silencing Dsg1 expression in keratinocytes increases the production of the melanocortins precursor *POMC* and increases melanin production and distribution even in the absence of UV stimulation [140]. Reinforcing the relevance of Dsg1 in melanocyte biology, Dsg1 is reduced in response to acute UV exposure [210]. Elevated levels of Tspan8 (Tetraspanin8) expression on the melanoma membrane confer the capacity to invade the dermis [211]. The mechanism of Tspan8-dependent dissolution of dermo-epidermal junction depends on the presence of keratinocytes and specifically on their capacity to produce MMPs and TIMPs [211]. Due to the tight correlation between Tspan8 expression and invasive capacity, Tspan8 has been proposed to predict metastasis risk in individual patients [212].

Melanoma cells lose the capacity to form gap junction communication with keratinocytes but still use these structures to communicate with themselves and with dermal fibroblasts [141]. Interestingly, several studies described the role of gap junctions in melanoma-immune cell interaction [213,214] indicating implications in anti-melanoma immune response and disease control. In skin cancer, Cx43 (connexin-43) is overexpressed in malignant melanomas when compared with the common and benign nevi, and higher levels of expression were found in patients with lymph node metastases. The upregulation of Cx43 was associated with tumor spreading since Cx43 mediates intracellular communication between the tumor microenvironment and the metastatic melanoma cells [215]. In contrast, abundant cytoplasmic Cx43 observed at later stages has been linked to invasion and metastasis [216]. Thus, the definition of oncogene has been proposed for Cx43 in melanoma; however, in human melanoma cell lines, Cx43 overexpression mitigates melanoma growth and metastasis and facilitates TNFα-induced cell apoptosis [217]. Cx31·1 and Cx26 are significantly downregulated in metastases compared with vertical growth phase melanomas. Cx26 expression in melanoma tissue has been shown to promote a metastatic cell phenotype and enhance the establishment of new tumor niches through cell-to-cell communication with the surrounding tissue. Very recently, Cx31 downregulation has been associated with melanoma metastasis, MAPK inhibitor resistance, and worse overall survival of patients [218].

The selectin family plays a major role in the progression of melanoma, as well as initiating complications in cancer patients such as coagulation defects, leading to a poorer prognosis [219]. Selectins are glycoproteins with a key role in cancer cell rolling, adhesion, extravasation, and the establishment of metastatic lesions [128]. E-selectin, (faintly expressed in normal melanocytes) has been shown to regulate the transepithelial migratory pathway through p38 and ERK kinases [219] and to facilitate immune evasion [128]. Melanocyte–keratinocyte physical interaction is not only critical to confine melanocytes in their physiological niche but also to maintain the regular protein surface profile of pigment cells. Accordingly, after isolation from skin normal human melanocytes in vitro express some melanoma-associated antigens [220].

### 4.2. Keratinocyte–Melanoma Cells Cross-Talk

Melanoma cells at the early stage are still fully embedded in the epidermis which is quite a homogenous environment. Thus, spontaneous as well as melanoma-driven alterations in keratinocyte layers undoubtedly serve for melanoma niche formation and progression. However, concerning the paracrine function of epidermal cells in melanoma, it is necessary to also take into consideration the role of intercellular communication with distant cells. Particularly, the complex melanocyte–keratinocyte–fibroblast network is critical for melanocyte homeostasis. Keratinocytes directly offer several mitogenic factors and indirectly stimulate the secretion of growth factors by fibroblasts. The acquisition of growth factor independence is a significant step in melanoma progression. Until the melanoma mutational profile does not involve key cell cycle regulators, the up modulation of growth factors by stromal cells appears to be the turning point for pro-tumorigenic switching of normal skin. While the hypersecretion of mitogenic molecules is a well-defined feature of CAFs, until now the corresponding secretory profile of melanoma-associated keratinocytes has not been fully elucidated. In wild-type BRAF melanocytes, the activation of the MAPK signaling can be because of abnormal production of mitogens by surrounding cells and by autocrine production of the growth factors bFGF and HGF [221,222]. Keratinocytes upregulate mitogenic factors in response to UV and during wound healing. Keratinocyte-derived NGF (nerve growth factor) serves to counteract UV-induced depletion of melanocytes by increasing the level of the anti-apoptotic BCL-2 protein [223], a mechanism that might help melanoma cells bypass critical phases. HGF, PDGF, and ET-1 decrease the expression of E-cadherin, leading to decoupling from keratinocytes [132,224]. Overall, these data evidence that E-cadherin is a very dynamic molecule that can relatively easily be down-modulated. Decoupling from keratinocytes does not necessarily reflect only E-cadherin levels in tumors since melanoma cells might orchestrate the remodeling of surface protein on keratinocytes. Emerging literature indicates that in addition to cell proliferation and survival, sphingolipid metabolism is implicated in the modulation of cell adhesion properties. Lipid mediators such as sphingosine 1-phosphate (S1P) released by melanoma cells reduce E-cadherin on the keratinocyte membrane and stimulate MMP9 expression in the same cells [225]. Correspondingly, melanoma-derived S1P activates stromal fibroblasts and promotes transdifferentiation into myofibroblasts [226].

The interaction of growth factors with proper receptors activates MAPK intracellular signaling and a cascade of events converging in cell division stimulation. However, chronic MAPK activation occurring in the case of BRAF mutations does not necessarily lead to loss of growth controls. Nevi harboring BRAF mutation with a frequency near 80% still require exogenous growth factors and have a finite life span similar to healthy melanocytes [63,221,222,227]. It is well-known that these precancerous focal proliferations of melanocytes stop growth due to the acquisition of oncogene-induced senescent features [228]. The exit of nevi from persistent cell senescence is paradigmatic in cancer biology. Using a microfluids system to leave intact indirect melanocyte/keratinocyte communication, a recent study showed that keratinocyte as well as dermal fibroblasts might alter the secretion of both pro- and anti-tumorigenic factors by transformed melanocytes leading to BRAF-induced senescence escape [229]. From the same study, emerged a phototype-dependent disparity in the secretome of melanocytes regardless of the neovascularization stimulation.

Expanding the forms of melanocyte–keratinocyte communication, Tagore et al. recently described a pro-melanomagenesis interaction based on GABAenergic (gamma-aminobutyrric acid) signaling [230]. Considering the embryological origin of melanocytes, it is not unexpected that GABA-synthesizing enzymes are expressed by pigment cells. However, the augmented level of GAD1 (glutamic acid decarboxylase) upon BRAFV600E activation is associated with a high level of GABA receptors in non-excitable keratinocytes directly adjacent to melanoma cells, suggesting an electrical activity-based communication is more surprising [230].

The contribution of epidermal keratinocytes to melanoma progression seems to be not restricted to cells residing in the proximity of melanocytic hyperprolipheration since in a very interesting investigation Golan and collaborators demonstrated that differentiated keratinocyte and not undifferentiated ones induce melanoma invasion via a Notch-dependent mechanism [231]. Confirming that keratinocytes are not unique static elements of the epidermis, when in contact with undifferentiated keratinocytes, melanocytes exhibit features of healthy melanocytes but when in contact with differentiated keratinocytes, melanocytes acquire the expression of phenotypic characteristics of nevus and melanoma cells [178].

Tumor lesions are frequently surrounded by persistent inflammation [232]. NALP3 (NRLR family pyrin domain containing 3) and additional components of the inflammasome multiprotein complex are highly activated in melanoma [233]. Inflammasome features correlate to a favorable prognosis in melanoma patients to the efficacy of immune checkpoint blockade therapy [233]. Local inflammatory mechanisms are associated with higher levels of circulating proinflammatory cytokines such as IL-8 (interleukin-8), IL-6, and INFα (interferon-α) [234]. Ultraviolet irradiation stimulates keratinocytes to secrete numerous cytokines that are responsible for the proliferation of melanocytes and transient melanogenesis activation. Chronic UVB results in IL-1, IL-8, and TNF (tumor necrosis factor) secretion by keratinocytes [235]. However, UV radiation is prevalently immune suppressive, a fact that might enable malignant cells to escape local (skin) immune control. Complicating the scenario, IL-12 in addition to the immune system regulatory function is involved in DNA damage repair of keratinocytes indicating that suppression of local immune response is necessary to restore compromised cells. Exposing the skin to solar-simulated UV radiation enhances the production of immunomodulatory lipids. Changes in lipid metabolism directly affect immune-cell phenotype and function, including increasing the production of cytokines that suppress the immune system. Interestingly, microvesicles generated by UVB-treated epidermal layers containing immunosuppressive molecules allow systemic effects [236]. Thus, body areas not exposed directly to UV radiation might receive immunosuppressive mediators derived from irradiated keratinocytes and released in blood circulation [237].

## 5. Conclusions

Rapidly dividing cells, such as keratinocytes, are quickly replaced by new differentiating cells due to the physiological turnover, whereas melanocytes are long-living cells constantly undergoing damage accumulation. Despite this significant difference, these two types of cells exist in a symbiotic relationship rarely perturbed during normal life. Skin regeneration, wound healing, and tanning response are tightly regulated processes that ensure the unmodified melanocyte/keratinocyte ratio. Also, cell adhesion features and direct contact between melanocytes and keratinocytes are only transiently modulated during skin remodeling. However, melanocytic hyperproliferation imposes a redesign of the epidermal picture. In the case of focal benign hyperproliferation such as in the case of nevi skin remodel reaches a stable point of equilibrium that rarely involves the dermis. By contrast, in the case of melanoma, tumor cells digress from the epidermis invading the dermis. This step is often associated with the switch from an epithelial-like to a mesenchymal-like phenotype of tumor cells. Thus, understanding the phenomena underlying the loss of epidermal homeostasis during the initial stages of melanoma is essential to optimize the diagnosis and treatment of early lesions.

## Figures and Tables

**Figure 1 cancers-16-00913-f001:**
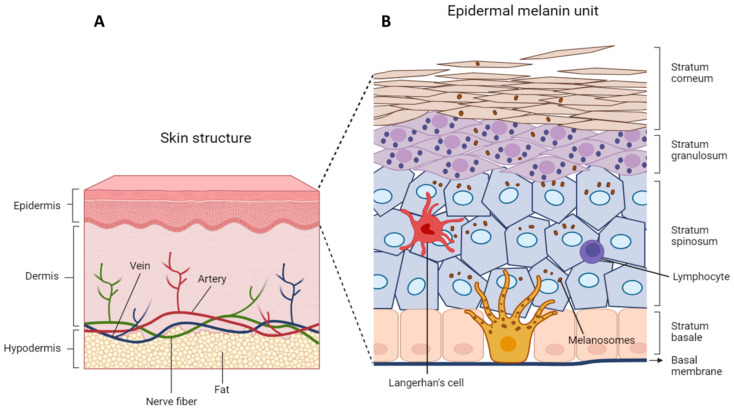
Schematic representation of skin and epidermal–melanin unit. (**A**) The skin structure is shaped in three layers: epidermis, the top layer; dermis, the middle layer; hypodermis, the bottom or fatty layer. The principal cells of the epidermis are keratinocytes at different levels of differentiation. The epidermis, composed of several connected cell sheets, is subdivided into four layers: the stratum basale (the deepest portion of the epidermis), stratum spinosum, stratum granulosum, stratum lucidum, and stratum corneum. Immediately below the epidermis is the basal membrane, a specialized structure that lies between the epidermis and dermis. The dermis is made of an abundant extracellular matrix composed of an interconnected mesh of elastin collagens produced by mesenchymal cells. The dermis also contains nerves and blood vessels. The hypodermis layer contains fat cells for energy reserve, additional connective tissue, and larger blood vessels. It gives elasticity and strength to the skin. (**B**) Details of the epidermal–melanin unit. The epidermal–melanin unit denotes the symbiotic relationship between a melanocyte and a group of associated keratinocytes. Melanocytes are highly dendritic, pigment-producing cells located in the deeper basal layer. The pigment is synthesized and packaged within specialized lysosome-derived organelles named melanosome and transferred to neighboring keratinocytes after movement across melanocyte dendrites. Keratinocytes accumulate melanin perinuclear as a cap around the nucleus. Within the epidermis, there are also sparse immune cells like Langerhans cells, Natural killer cells, and lymphocytes.

**Figure 2 cancers-16-00913-f002:**
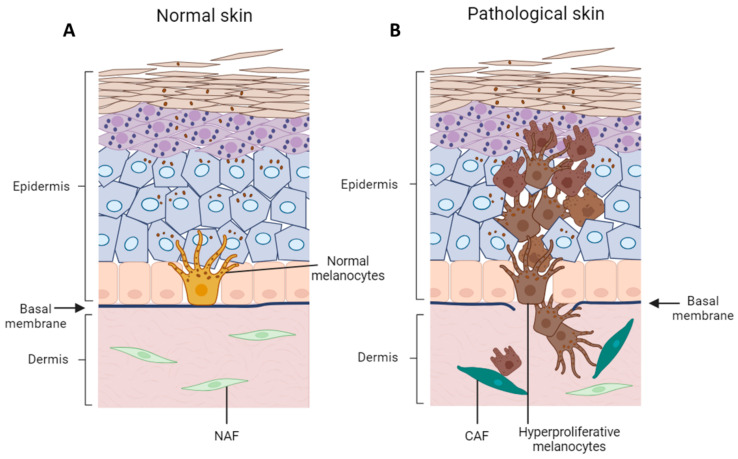
Cell interaction in normal versus pathological skin. (**A**) In healthy skin, epidermal cells are tightly interconnected. Through their dendrites, melanocytes interact with about 36–40 keratinocytes. The integral basal membrane preserves the separation of dermal and epidermal compartments. The dermis is made of abundant extracellular matrix produced by normal tissue-associated fibroblasts (NAF). (**B**) In pathological skin, for example during melanoma development, a loss of connections takes place between melanocytes with keratinocytes and basal membrane. During melanocytic abnormal proliferation, the entire cutis undergoes progressive structural and functional rearrangement. If this process is limited to a group of cells the ultimate lesion consists of nevomelanocytes presenting features of senescent cells. Normal melanocyte mutates into melanoma cells in a multi-step process. The uncontrolled horizontal or radial growth phase is the first step towards the invasive phenotype, in which melanocytes display a hyperproliferative phenotype and undergo alterations that offer a survival advantage. This is followed by a vertical growth phase, in which tumor cells deeply invade the dermis. This step implies damage to the basal membrane. Hyperproliferative melanocytic cells escape keratinocyte control and instead interact with fibroblasts and other hyperproliferative melanocytic cells. Melanoma progressively shapes the surrounding microenvironment in a more favorable milieu, a process that is mostly due to the transformation of NAF (light green) in cancer-associated fibroblasts CAF (dark green).

**Figure 3 cancers-16-00913-f003:**
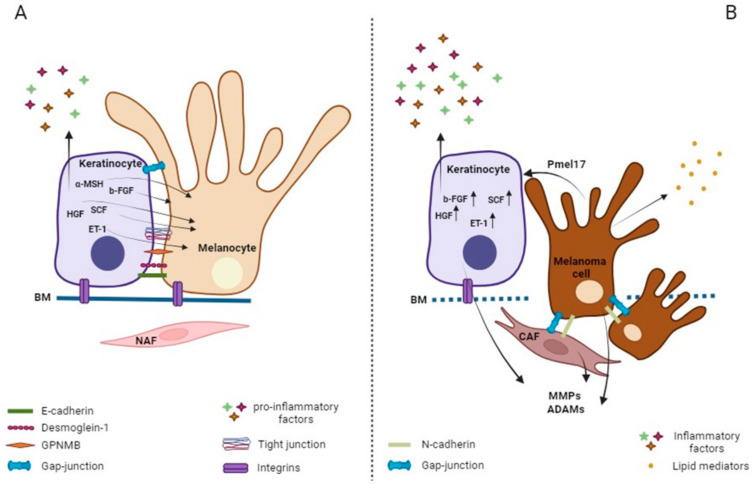
Concise representation of melanocyte–keratinocyte interaction during melanomagenesis. The release of several soluble factors normally released by keratinocytes for paracrine functions (**A**) are up-regulated during melanomagenesis (**B**). Among these factors, mitogens and inflammatory mediators play the most relevant role. MMPs, ADAMs and TIMPs secreted by melanoma and stromal cells are responsible of tissue remodeling including basal membrane (BM) invasion and the entrance in the dermis where fibroblasts assume CAF phenotype.

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
