# Peer review of "The Keratinocyte in the Picture Cutaneous Melanoma Microenvironment"

_cancers, 2024, doi:10.3390/cancers16050913_

Round 1

Reviewer 1 Report

Comments and Suggestions for Authors

The manuscript geves a comprehensive picture about  keratinocyte in the cutaneous melanoma microenvironment. Well organized, clearly written. The figures are well structured. It is an important review, containg many relevant references of the field.

Comment:

Figure 1: text, I would correct as: Schematic representation of skin structure and epidermal-melanin unit

Figure 2. The labels (A and B) are missing from the figure.

The letter size are different in some sentemces e.g. page 12 (Jun, c-Myc and CyclinD-1); r POMC

Author Response

Editorial Office

Cancers                                                                                                                 February 8th, 2024

Revised manuscript cancers-2860912

Dear Editor,

please find enclosed the revised version of the invited Review (R3) article entitled “The keratinocyte in the picture cutaneous melanoma microenvironment by Ramona Marrapodi and Barbara Bellei. We have carefully considered all comments and we have revised the manuscript accordingly to the requests and answered the comments point-by-point as follows:

Reviewer 1

The manuscript geves a comprehensive picture about keratinocyte in the cutaneous melanoma microenvironment. Well organized, clearly written. The figures are well structured. It is an important review, containg many relevant references of the field.

Comment:

Figure 1: text, I would correct as: Schematic representation of skin structure and epidermal-melanin unit

The figure legend has been modified.

Figure 2. The labels (A and B) are missing from the figure.

We apologize, the Figure 2 uploaded was not the final version and consequently, we missed some details. In the revised manuscript it was replaced by the correct one.

The letter size are different in some sentemces e.g. page 12 (Jun, c-Myc and CyclinD-1); r POMC

The entire text has been corrected.

Reviewer 2 Report

Comments and Suggestions for Authors

This is a updated review about the cellular processes related to the keratinocyte-melanocyte interactions during the normal and malignant transformation of melanoma in the skin. It is written from an integral point of view. The complexity of the processes is illustrated by mentioning the great number of gene and protein factors that are involved in the process. The review is well organized, and the number of perfectly referenced. In my opinion, this review would need some more figures or schemes about the main intracellular reactions and signal proteins related to the interactions among those types of cells and the differences between keratinocytes and normal or malignant melanocytes. The two figures of the review are quite simple in comparison with the content of the article. Anyway,  it is a brilliant review, clear in spite of the enumeration of so many signal transductions systems and proteins involved in the cross-talk of keratinocytes, melanocytes, dermal fibroblasts. It should be accepted for publication once the following minor points would be addressed.

1)   Unification of the letter size according to the format specified in the instructions to authors. See for instance the words “corneocyte lipid envelope” at page 6, or cytophagocytosis at page 7. There are more examples, for instance a paragraph at page 12.  In other cases, the size if identical, but the letter style is not uniform. For instance, at page 11… Cell adhesion molecules including cadherin, integrin, immunoglobulin, sometimes calibri, sometimes palatino.

2)   The number of the melanosomal proteins would be unified. For instance, Pmel17 is mentioned several times before the protein would be considered gp100/Pmel17. This protein is also named silver protein as an old denomination observed in the color of mice hair mutated in the PMEL locus. The names gpxx is not recommended (for instance, TRP1 was formerly named gp75).

3)   The MMP involved in the process: In one paragraph, It is mentioned MMP9 and MMP2, but in other, there are more (1,2,9 and 19). Does it mean that MMP1 and MMP19 are related to melanomagenesis but MMP2 and MMP9 to just inflammation?. I think this is not correct, but I would like some clarification.

Page 3. One paragraph starts with the sentence “MC1R gene favors a DNA damage response…”. This is mostly because MC1R favors the synthesis of pheomelanin, and the light-absorbing capacity of pheomelanin is much lower than eumelanin. In fact, it has been demonstrated that pheomelanin is more a photosensitizer than a photoprotector. This point would be mentioned to complete the paragraph.

Page 5. At the end of a paragraph containing the sentence “Lastly, adipose tissue-derived cells are a local source of CAF [84]”. Further comments are needed for this sentence. To me, it is not obvious that these cells are source of Cancer Associated Fibroblasts and that these cells are needed for melanoma development (as it is established at Figure 2). It seems to me that melanoma can be produced by direct malignancy of melanocytes. The role of CAF cells is not clear to me. I would ask to the authors the clarification of this point.

Page 7, line 3: The transportation of “such specialized lysosome-derived organelles”. These organelles are melanosomes, but the name should be mentioned before the sentence (not later on).

Page 7: Sentence “Tyrosine, a nonessential amino acid, serves as the precursor molecule for the production of all types of melanins”. This is true, but cysteine (or glutathione) should be cited as the extra amino acid needed for pheomelanin. Otherwise, cysteinyldopa cannot be formed.

End of Page 7/head of Page 8: Physiological keratinocyte-melanocyte communication. The initial paragraph is a repetition, and it would be deleted. By the way, recent data indicate that the number of keratinocytes in the epidermal unit is probably lower than 36-40. Some authors decrease the number to aprox. 12. In fact, it depends on the skin region. A sentence about this variability would be welcome.

Page 11: Delete the letter f in the expression “of f nevomelanocytes”.

Author Response

Editorial Office

Cancers                                                                                                                 February 8th, 2024

Revised manuscript cancers-2860912

Dear Editor,

please find enclosed the revised version of the invited Review (R3) article entitled “The keratinocyte in the picture cutaneous melanoma microenvironment by Ramona Marrapodi and Barbara Bellei. We have carefully considered all comments and we have revised the manuscript accordingly to the requests and answered the comments point-by-point as follows:

Reviewer 2

This is a updated review about the cellular processes related to the keratinocyte-melanocyte interactions during the normal and malignant transformation of melanoma in the skin. It is written from an integral point of view. The complexity of the processes is illustrated by mentioning the great number of gene and protein factors that are involved in the process. The review is well organized, and the number of perfectly referenced. In my opinion, this review would need some more figures or schemes about the main intracellular reactions and signal proteins related to the interactions among those types of cells and the differences between keratinocytes and normal or malignant melanocytes. The two figures of the review are quite simple in comparison with the content of the article. Anyway, it is a brilliant review, clear in spite of the enumeration of so many signal transductions systems and proteins involved in the cross-talk of keratinocytes, melanocytes, dermal fibroblasts. It should be accepted for publication once the following minor points would be addressed.

As suggested, we prepared an additional figure for a graphical representation of details regarding the keratinocyte-melanocyte crosstalk.

  • Unification of the letter size according to the format specified in the instructions to authors. See for instance the words “corneocyte lipid envelope” at page 6, or cytophagocytosis at page 7. There are more examples, for instance a paragraph at page 12.  In other cases, the size if identical, but the letter style is not uniform. For instance, at page 11… Cell adhesion molecules including cadherin, integrin, immunoglobulin, sometimes calibri, sometimes palatino.

We apologize for the disarray. In the revised manuscript we used the Calibri 12 for the word text.

  • The number of the melanosomal proteins would be unified. For instance, Pmel17 is mentioned several times before the protein would be considered gp100/Pmel17. This protein is also named silver protein as an old denomination observed in the color of mice hair mutated in the PMEL locus. The names gpxx is not recommended (for instance, TRP1 was formerly named gp75).

The text has been revised to correctly introduce the proteins with the names used throughout the document and indicate any synonyms used in the literature.

  • The MMP involved in the process: In one paragraph, It is mentioned MMP9 and MMP2, but in other, there are more (1,2,9 and 19). Does it mean that MMP1 and MMP19 are related to melanomagenesis but MMP2 and MMP9 to just inflammation?. I think this is not correct, but I would like some clarification.

We agree with the reviewer, this part has been too briefly reported. The intention was to discuss some non-oncological aspects of MMPs function in skin homeostasis, also including some pathological but non-oncological aspects. With this aim, we have revised the text explaining more extensively the role of UV and inflammation in the regulation of proteins of the MMPs and ADAMs families. The part referring to the role of these proteins in tumor progression has been expanded on page 19.

Page 3. One paragraph starts with the sentence “MC1R gene favors a DNA damage response…”. This is mostly because MC1R favors the synthesis of pheomelanin, and the light-absorbing capacity of pheomelanin is much lower than eumelanin. In fact, it has been demonstrated that pheomelanin is more a photosensitizer than a photoprotector. This point would be mentioned to complete the paragraph.

The lower light-absorbing capacity of red melanin has been reported on page 4 and it is not generally referred to as MC1R but to the polymorphic variants linked to the prevalent synthesis of reddish pheomelanin having photosensitizer properties.

Regarding the sentence reported by the reviewer “MC1R gene favors a DNA damage response……………..…” it is cited to discuss the DNA damage response triggered by MC1R engagement by its ligand (10.1111/pcmr.12823  10.1158/1541-7786.MCR-11-0436  10.1111/exd.12420). MC1R signals increase the capacity to tolerate all environmental stressors. Also this function is impaired in MC1R variants and consequently, the somatic mutation burden in melanoma from subjects with MC1R variants is higher compared to individuals with the wild-type sequence. We reported better these data extending the discussion in a few more lanes.

Page 5. At the end of a paragraph containing the sentence “Lastly, adipose tissue-derived cells are a local source of CAF [84]”. Further comments are needed for this sentence. To me, it is not obvious that these cells are source of Cancer Associated Fibroblasts and that these cells are needed for melanoma development (as it is established at Figure 2). It seems to me that melanoma can be produced by direct malignancy of melanocytes. The role of CAF cells is not clear to me. I would ask to the authors the clarification of this point.

The sentence at the end of the paragraph summarized the following concept:

Cancer associated fibroblasts (CAFs) play key roles in regulating the biologic function of the tumor stroma and contribute to immune regulation, angiogenesis, and ECM remodeling of the tumor, as well as the generation and maintenance of cancer stem cells, thereby promoting therapeutic resistance. CAFs are defined by a combination of their morphology, association with cancer cells, and lack of lineage markers for epithelial cells, endothelial cells, and hematopoietic cells (10.1038/s41568-019-0238-1; 10.3390/cancers14163906). CAFs often develop from local resident fibroblast populations but can also differentiate from mesenchymal stromal cells or mesenchymal stem cells. Further, outside of the fibroblast lineage, CAFs can transdifferentiate from epithelial cells, blood vessels, adipocytes, pericytes, and smooth muscle. Specifically, it has been demonstrated especially in breast cancer that adipocytes phenotypically change to generate fibroblast-like cells termed adipose-derived fibroblasts expressing some specific CAF markers such as FSP1 and a marked pro-inflammatory phenotype (10.1158/0008-5472.CAN-13-0530; 10.1007/s13277-015-3594-9).

This aspect is also presented at the end of the introduction. However, we extended this concept in the revised text. We did not modified the figure since in this case the important consideration is the presence of CAF but not the exact origin of this stromal component. In any way the authors well known that CAFs are prevalently originated by fibroblasts but at the same time it is important to mention also additional possible mechanism.

Page 7, line 3: The transportation of “such specialized lysosome-derived organelles”. These organelles are melanosomes, but the name should be mentioned before the sentence (not later on).

The sentence has been corrected.

Page 7: Sentence “Tyrosine, a nonessential amino acid, serves as the precursor molecule for the production of all types of melanins”. This is true, but cysteine (or glutathione) should be cited as the extra amino acid needed for pheomelanin. Otherwise, cysteinyldopa cannot be formed.

As requested, this point has been clarified.

End of Page 7/head of Page 8: Physiological keratinocyte-melanocyte communication. The initial paragraph is a repetition, and it would be deleted. By the way, recent data indicate that the number of keratinocytes in the epidermal unit is probably lower than 36-40. Some authors decrease the number to aprox. 12. In fact, it depends on the skin region. A sentence about this variability would be welcome.

As requested, we included different data reporting different considerations about the melanocyte distribution tacking also in account body area and age (pag.12).

Page 11: Delete the letter f in the expression “of nevomelanocytes”.

It has been corrected.

Reviewer 3 Report

Comments and Suggestions for Authors

A review paper by Ramona Marrapodi and Barbara Bellei discusses remodeling that occurs in the epidermis during melanoma formation as well as skin changes that occur independently to melanocytic hyperproliferation having possible pro-tumoral features. The review is written comprehensively, and touches a novel aspects of the interactions between keratinocytes and melanoma.

Specific comments:

1. English should be revised, e.g., in abstract "This review will discusses remodeling that occur" should be "This review will discuss remodeling that occurs" etc.

2. Several papers in the field are missing, e.g., 

doi: 10.1016/j.celrep.2023.113586

doi: 10.1093/bjd/ljad474

doi: 10.1158/2159-8290.CD-23-0843

Comments on the Quality of English Language

requires revision

Author Response

Editorial Office

Cancers                                                                                                                 February 8th, 2024

Revised manuscript cancers-2860912

Dear Editor,

please find enclosed the revised version of the invited Review (R3) article entitled “The keratinocyte in the picture cutaneous melanoma microenvironment by Ramona Marrapodi and Barbara Bellei. We have carefully considered all comments and we have revised the manuscript accordingly to the requests and answered the comments point-by-point as follows:

Reviewer 3

A review paper by Ramona Marrapodi and Barbara Bellei discusses remodeling that occurs in the epidermis during melanoma formation as well as skin changes that occur independently to melanocytic hyperproliferation having possible pro-tumoral features. The review is written comprehensively, and touches a novel aspects of the interactions between keratinocytes and melanoma.

Specific comments:

  1. English should be revised, e.g., in abstract "This review will discusses remodeling that occur" should be "This review will discuss remodeling that occurs" etc.

As suggested, we fully revised the manuscript.

  1. Several papers in the field are missing, e.g., 

We would like to thanks the Reviewer for the proposal of these interesting publications. We carefully analyzed data included and we used some information to amply our review.

Specifically, doi: 10.1016/j.celrep.2023.113586 has been largely discuss at pag. 21;

doi: 10.1093/bjd/ljad474 has been included in the introduction at page 3;

doi: 10.1158/2159-8290.CD-23-0843 Regarding this article, we preferred report the original reference rather than the suggested commentary. This part has been inserted at pag.22 of the revised manuscript.
